# Associations between Greenspace and Gentrification-Related Sociodemographic and Housing Cost Changes in Major Metropolitan Areas across the United States

**DOI:** 10.3390/ijerph18063315

**Published:** 2021-03-23

**Authors:** Leah H. Schinasi, Helen V. S. Cole, Jana A. Hirsch, Ghassan B. Hamra, Pedro Gullon, Felicia Bayer, Steven J. Melly, Kathryn M. Neckerman, Jane E. Clougherty, Gina S. Lovasi

**Affiliations:** 1Department of Environmental and Occupational Health, Dornsife School of Public Health, Drexel University, Philadelphia, PA 19104, USA; jec373@drexel.edu; 2Urban Health Collaborative, Dornsife School of Public Health, Drexel University, Philadelphia, PA 19104, USA; Jah474@drexel.edu (J.A.H.); Fjb47@drexel.edu (F.B.); sjm389@drexel.edu (S.J.M.); gsl45@drexel.edu (G.S.L.); 3Medical Research Institute of the Hospital del Mar (IMIM), 08003 Barcelona, Spain; helen.cole@uab.cat; 4Institute for Environmental Science and Technology, Universidad Autónoma de Barcelona, 08193 Bellaterra, Spain; 5Department of Epidemiology & Biostatistics, Dornsife School of Public Health, Drexel University, Philadelphia, PA 19104, USA; 6Department of Epidemiology, Johns Hopkins Bloomberg School of Public Health, Baltimore, MD 21205, USA; ghassanhamra@jhu.edu (G.B.H.); Pedro.gullon@edu.uah.es (P.G.); 7Public Health and Epidemiology Research Group, School of Medicine and Health Sciences, Universidad de Alcala, Alcala de Henares, 28801 Madrid, Spain; 8Columbia Population Research Center, Columbia University, New York, NY 10027, USA; kmn2@columbia.edu

**Keywords:** gentrification, green, greenspace, spatial, socioeconomic position, income, race, poverty, urban

## Abstract

Neighborhood greenspace may attract new residents and lead to sociodemographic or housing cost changes. We estimated relationships between greenspace and gentrification-related changes in the 43 largest metropolitan statistical areas (MSAs) of the United States (US). We used the US National Land Cover and Brown University Longitudinal Tracts databases, as well as spatial lag models, to estimate census tract-level associations between percentage greenspace (years 1990, 2000) and subsequent changes (1990–2000, 2000–2010) in percentage college-educated, percentage working professional jobs, race/ethnic composition, household income, percentage living in poverty, household rent, and home value. We also investigated effect modification by racial/ethnic composition. We ran models for each MSA and time period and used random-effects meta-analyses to derive summary estimates for each period. Estimates were modest in magnitude and heterogeneous across MSAs. After adjusting for census-tract level population density in 1990, compared to tracts with low percentage greenspace in 1992 (defined as ≤50th percentile of the MSA-specific distribution in 1992), those with high percentage greenspace (defined as >75th percentile of the MSA-specific distribution) experienced higher 1990–2000 increases in percentage of the employed civilian aged 16+ population working professional jobs (β: 0.18, 95% confidence interval (CI): 0.11, 0.26) and in median household income (β: 0.23, 95% CI: 0.15, 0.31). Adjusted estimates for the 2000–2010 period were near the null. We did not observe evidence of effect modification by race/ethnic composition. We observed evidence of modest associations between greenspace and gentrification trends. Further research is needed to explore reasons for heterogeneity and to quantify health implications.

## 1. Introduction

A growing body of literature demonstrates links between greenspace and improved health endpoints [1], including lower rates of all-cause [2], respiratory [3], circulatory disease [4], and infant [5] mortality. Indeed, greenspace may contribute to improvements in physical activity [6] and mental health [7] via several possible mechanisms, including improved social cohesion [8], reduced noise [9], and violent crime rates [10], as well as reduced urban heat islands [11]. Despite this growing evidence base, there remain calls for more and improved quantitative population health research to empirically evaluate and document the health benefits of greenspace [12]. These calls are motivated by a number of knowledge gaps and limitations in the research conducted to date, including lack of clarity on the mechanisms via which greenspace improves health, concerns that past research on links between greenspace and health are biased due to confounding by socioeconomic position, and worries that before vs. after greenspace installation comparisons may be biased as a result of wealthier (and potentially healthier) residents self-selecting themselves into areas that have been greened [12]. The issue of self-selection of higher-socioeconomic-position residents into greener areas is a substantive concern, in itself, as it implies the potential exclusion—either via physical displacement or through cultural or social isolation—of lower-socioeconomic-position residents, and it suggests that new or improved green spaces in cities may amplify inequities in health and quality of life [13,14].

Indeed, previous studies have found that the production or restoration of parks or green amenities can lead to rising housing costs and sociodemographic changes that are consistent with gentrification [15,16,17,18], which we define here as the repopulation of previously disinvested sections of cities by middle- and upper-class residents [19]. From the perspective of community health promotion, the possibility that greenspace may contribute to gentrification processes is a critical concern and suggests a potential failure, on some levels, of green programs from achieving their intended goal of improving community well-being [20]. Indeed, empirical research has shown that gentrification may intensify health inequities, simultaneously having largely negative impacts on the health of racial and ethnic minorities and low-socioeconomic-position populations, but protective or no impacts on the population at large or among more privileged subgroups [21,22,23].

Recently, scholars have highlighted the need for further investigation of links between greened or protected areas and gentrification [24]. Despite this interest, few studies have quantified relationships between greenspace and gentrification-related neighborhood change processes across multiple urban areas. A large-scale, nationwide analysis is needed to allow comparison of trends and relationships across areas, and it may help inform the design, conduct, and interpretation of results from large, population-based analyses that estimate associations between greenspace and health.

To address this gap, we analyzed relationships between census tract-level percentage greenspace and gentrification-related neighborhood changes in major metropolitan areas across the United States (US). Our underlying hypothesis was that the presence of greenspace (both newly produced and pre-existing) encourages gentrification processes. We also hypothesized that there would be heterogeneity in these associations across areas. Such heterogeneity could result from different population characteristics, spatial correlation of greenspace with other amenities that attract gentrification, or variations in local greening strategies (e.g., contrasting so-called “just green enough” strategies in which green spaces are designed with the idea of promoting community interests, with green development strategies that serve the local economic markets [20]). Our analytic goals were to quantify associations between baseline greenspace and subsequent neighborhood compositional changes, as well as to quantify heterogeneity in these relationships across major US cities. We also quantified the extent to which associations differed on the basis of racial composition of the census tracts. This work contributes to the scholarly literature on associations between greenspace and gentrification [15,17,18] by using a uniform, nationwide greenspace measure to quantify associations across multiple major metropolitan areas across the US and by considering relationships with any greenspace type, including pre-existing and natural greenspace (e.g., forests), as well as newly created parks.

## 2. Materials and Methods

### 2.1. Study Design

Our analytic goal was to quantify the extent to which baseline percentage greenspace was associated with subsequent neighborhood changes. To achieve this goal, we conducted a longitudinal, ecologic analysis of associations between percentage greenspace in either 1990 or 2000 and subsequent neighborhood changes in the periods 1990–2000 and 2010, respectively, in major metropolitan areas across the United States. These time periods were selected on the basis of data availability and because we hypothesized that a 10 year time span was sufficient to observe neighborhood changes. The unit of analysis was the census tract, clustered within metropolitan statistical areas (MSAs). We used a two-stage, hierarchical approach, firstly estimating associations in each MSA, and secondly pooling the estimates in a meta-regression to derive summary estimates.

We restricted the analysis to all MSAs in the conterminous US containing 300 or more census tracts (*N* = 43). MSAs are delineated by the US Office of Management and Budget (OMB) and defined as one or more counties containing (a) at least one city with a population of 50,000 or more, or (b) an urbanized area (defined by the Census Bureau) with a total population of at least 100,000 [25]. Additional adjacent counties may be included in the MSA if they have high amounts of social and economic integration because of commuting activities.

### 2.2. Data

We used 2010 census tract boundaries to approximate neighborhoods. We derived data on census tract-level population densities, sociodemographics, and housing costs (described in detail below) from the Brown University Longitudinal Tracts database (LTDB) [26]. The LTDB provides estimates of census tract-level variables, all based on 2010 boundaries. Variables for years 1990, 2000, and 2010 were derived from each Decennial Census and from 2008–2012 American Community Survey 5 year estimates.

#### 2.2.1. Greenspace Measure

We estimated percentage greenspace in each census tract using data from the Multiresolution Land Characteristics (MRLC) consortium’s National Land Cover Database (NLCD) [27,28]. The NLCD provides land-cover data for the entire US at a 30 m resolution. It uses a 16-class legend based on a modified Anderson Level classification system. We calculated percentage greenspace in each census tract as the total area in each census tract classified as green (forested upland, shrubland, non-natural woody, and herbaceous upland/seminatural vegetation; Modified Anderson Level 1 class codes 4–7) divided by the total land area of each census tract [29].

We used greenspace data from the closest available years to years 1990 and 2000. Specifically, we used NLCD from 1992 to approximate percentage greenspace in 1990 and NLCD from 2001 to approximate percentage greenspace in 2000. We assumed that changes in greenspace occur slowly and, therefore, that estimates of percentage greenspace in 1992 and 2001 adequately represent coverage in years 1990 and 2000, respectively. The MRLC consortium changed their methods for assessing land cover between years 1992 and 2001 [29]. In both years, the data were created using Landsat TM imagery of the conterminous US. In 1992, the consortium used an unsupervised classification system to create spectral classes for the Landsat TM images and a hybrid classification system to classify the landcover types. In 2001, the consortium classified images using a decision tree method, rather than the unsupervised and hybrid systems. Because of these changes, the data are internally valid but cannot be used to compare percentage greenspace between 1992 and 2001. For this reason, we avoid directly comparing the magnitude of the quantitative estimates of association between percentage greenspace in either 1992 or 2001 and subsequent neighborhood changes.

#### 2.2.2. Gentrification Measure

We estimated associations with one categorical variable representing gentrification (gentrified vs. did not gentrify) and with nine gentrification-related sociodemographic or housing cost change variables. There is no consensus on how to define gentrification. Generally, the term describes the reversal of inner-city disinvestments in urban neighborhoods and the resultant influx of middle- to upper-class residents [30]. Gentrification is commonly equated with or assumed to result in the displacement of long-term residents in previously disinvested areas; however, empirical studies have shown that displacement is not an inevitability [19,30,31].

As with its definition, there is no consensus on optimal methods to measure gentrification [32]. A variety of methods and data sources have been used previously to quantify gentrification. These include using census data to characterize area-level sociodemographic shifts [30], using home loan data [33], or characterizing physical changes to the built or retail environments that may indicate neighborhood shifts (e.g., coffee shops) [34,35]. Many scholars use a two-step approach to identify gentrification [19,30]. The first step is to identify neighborhoods (often defined using census tracts, for convenience) that are “gentrifiable” at baseline, i.e., areas that are disinvested and/or low-income and are, therefore, available for upgrading and revitalization. The second step is to identify neighborhoods that experienced gentrification—from among those that were determined to be eligible to gentrify. Being classified as having gentrified or not is often based on evidence that the neighborhood experienced (a) sociodemographic shifts that indicate an influx of persons of higher education, income, or wealth, and (b) reinvestment, identified on the basis of increases in real estate costs. Numerous census-based measures, including educational attainment, home sales price, home rent price, professional work, and racial/ethnic composition [14,36], have been used to quantify these changes. Often, cut-points used to characterize whether sociodemographic shifts are substantial enough to represent gentrification are based on the magnitude of such changes relative to the larger region to which the neighborhood belongs (e.g., above the median increases compared to the metropolitan area) [19].

Our approach to operationalizing gentrification followed the two-step approaches used by others. As others have done, we used a pre-specified cut-point to identify gentrifiable areas [16,19,30]. Then, among the census tracts eligible to gentrify, we used two methods to identify and characterize gentrification. First, in preliminary analyses, we classified census tracts into the following categories: (1) eligible, did not gentrify and (2) eligible, gentrified. Then, recognizing that gentrification is a complex process, and because we were interested in understanding the dimensions of neighborhood change that neighborhoods experienced, we quantified associations between baseline percentage greenspace and changes in sociodemographic or housing cost variables. This approach allowed us to characterize the type, extent, and directionality of neighborhood change processes, and it did not require placing census tracts into discrete categories (gentrified vs. did not gentrify). We identified gentrifiable census tracts and assessed neighborhood change processes for the decades 1990–2000 and 2000–2010 separately. That is, period one (1990–2000) was considered in isolation from period two (2000–2010). In the sections below, we describe these steps in detail.

Step one: Identifying tracts that were eligible to gentrify. In step one, we excluded tracts with fewer than 50 people in the baseline period (either 1990 or 2000). We did this in order to restrict our analyses to residential census tracts. Among the remaining tracts, we identified a census tract as “eligible to gentrify” if it was below the top quartile of median household income at baseline (1990 or 2000). As in most previous gentrification analyses, we defined eligibility by comparing each census tract to the distribution within the MSA to which it belonged.

Step two A: Categorical variable defining census tracts as gentrified or not. Among the tracts that were eligible to gentrify, we identified gentrified tracts as those that experienced an above-median increase in percentage of residents with a bachelor’s degree (or higher) and either (1) an above-median increase in median monthly contract rent (inflation-adjusted), or (2) an above-median increase in home value. These variables were selected because they represent changes in sociodemographics and real estate costs, and because they have been used in several previous analyses of gentrification [19,30]. As with previous measures, all metrics for a given census tract were compared to the distributions in their MSA.

Step two B: Gentrification-related sociodemographic or housing cost changes. Among tracts that were eligible to gentrify, we estimated associations with changes in the following nine tract-level measures of sociodemographic composition or real-estate prices that are commonly associated with gentrification: percentage of the census tract population living below the poverty line, percentage non-Hispanic Black, percentage non-Hispanic white, percentage Hispanic, percentage of the tract’s employed civilian population ages 16 and up working professional jobs, percentage of the census tract population ages 25 and older with a college education, inflation-adjusted median home value, inflation-adjusted median household income, and inflation-adjusted median monthly contract rent. For the 1990–2000 analysis, we quantified changes by subtracting the census tract values in 1990 from those in 2000. For the 2000–2010 analysis, we subtracted the values in 2000 from those in 2010.

### 2.3. Statistical Analyses

#### 2.3.1. Overview

First, as a *preliminary analysis*, we estimated associations between percentage greenspace and the categorical gentrification measure. Second, as a *main analysis*, we quantified associations between percentage greenspace and changes over each decade in sociodemographics or housing costs. For both analyses, we ran separate models for each MSA, and then we combined the MSA-specific estimates using random-effects meta-analysis. We used this hierarchical approach to allow computation of individual estimates for each MSA, hypothesizing that associations vary from place to place. Descriptive statistics on the gentrifiable census tracts within each MSA are given in Appendix A (Appendix A).

In all sets of models, we coded percentage greenspace on the basis of each MSA-specific distribution. In the 1990–2000 period models, we coded percentage greenspace as a three-level categorical variable (0–50th percentile (referent), >50th percentile–75th percentile, and >75th percentile) as a function of the percentage greenspace distribution in each MSA in 1992. In the 2000–2010 period models, we coded percentage greenspace as a two-level categorical variable (0–75th percentile (referent) and >75th percentile) as a function of the percentage greenspace distribution in each MSA in 2001. We used three categories for the earlier time period in order to observe dose–response relationships. We used two rather than three categories for the later time period because, in some MSAs, the median of the distribution for percentage greenspace was zero. The MSA-specific distributions for the two time periods are given in Appendix A (Appendix A).

#### 2.3.2. Preliminary Analyses

We used generalized linear models to quantify associations between (1) percentage greenspace in 1992 and gentrification from 1990–2000, and (2) percentage greenspace in 2001 and gentrification from 2000–2010. We used a modified Poisson regression approach with a robust error variance because of convergence issues when we used a log-binomial distribution, and because the outcome was common, meaning that odds ratios from logistic models would be systematically further from the null than the corresponding risk ratios [37].

#### 2.3.3. Main Analyses

We observed evidence of strong positive spatial autocorrelation in the greenspace measures, according to the Global Moran’s Index (*p* < 0.001 for percentage greenspace in 1992 and 2001; Appendix A). Therefore, for our main analyses, we used spatial simultaneous autoregressive lag models to quantify associations between (1) percentage greenspace cover in 1992 and neighborhood change from years 1990–2000, and (2) percentage greenspace in 2001 and neighborhood change from years 2000–2010. We used the scale function in R to *z*-score standardize the sociodemographic and housing cost change variables [38]. Spatial lag models assume that the values of the dependent variable in one census tract are associated with neighboring census tracts and include a spatially lagged dependent variable as an additional predictor. We used a first-order, queen contiguity matrix to define neighbors, meaning a census tract was identified as a neighbor if at least part of its boundary or vertice was shared by another census tract [39].

#### 2.3.4. Adjustment for Population Density

After running both the preliminary and the main analyses using univariate models, we repeated them with adjustment for census tract-level population density in 1990 in the model with percentage greenspace in 1992 as the independent variable and in 2000 in the model with percentage greenspace in 2001 as the independent variable. To accommodate potential nonlinearity of the associations, we coded population density as a four-level categorical variable on the basis of MSA-specific quartiles of the distribution for the relevant year. We adjusted for population density as a potential confounder because baseline population density levels may be negatively correlated with percentage greenspace (i.e., lower percentage greenspace in more population-dense/urban areas) and either positively or negatively related to urbanization and, thus, gentrification trends. For example, more population-dense/urban areas may be subject to gentrification trends because of competing attractions, aside from greenspace. Alternatively, less population-dense areas may be negatively related to gentrification processes because they offer space for development.

#### 2.3.5. Effect Measure Modification

We explored percentage non-Hispanic Black and percentage Hispanic, in 1990 for the 1990–2000 period models, and in 2000 for the 2000–2010 period models, as potential effect measure modifiers by including interaction terms between percentage greenspace and each of the race/ethnic composition variables. We ran separate models for each potential effect measure modifier. In the interaction models, for simplicity, we coded percentage greenspace in both periods as two-level categorical terms (0–75th percentile of the MSA-specific distribution (referent) vs. >75th percentile). We coded the effect measure modifier terms as two-level categorical variables (0–50th percentile in each MSA vs. >50th percentile). We selected the 50th percentile of the race/ethnic variables as the cut-point to sufficiently accommodate high vs. low composition comparisons and to ensure that, in each MSA, there were sufficient numbers of census tracts falling into each of the contrast categories of interest.

### 2.4. Sensitivity Analyses

To test the robustness of our results, we conducted several sensitivity analyses. First, we reran unadjusted spatial lag models using a first-order rook contiguity matrix to define neighbors, meaning tracts that shared common boundaries were defined as neighbors [39]. We also reran unadjusted models: (1) for all census tracts, regardless of whether they were eligible to gentrify, and (2) defining as eligible to gentrify all census tracts that were below the 50th percentile (rather than below the top quartile) of median household income in either 1990 or 2000.

### 2.5. Reporting

We report all the meta-analytic summary estimates from the preliminary analyses as risk ratios and 95% confidence intervals (CIs). We report meta-analytic summary estimates from the main analyses as β coefficients and 95% CIs. We also report the I^2^ statistic from the meta-analyses as an indicator of heterogeneity in associations across MSAs. This statistic represents the percentage of the variation across MSAs that is due to heterogeneity rather than chance; larger values represent greater heterogeneity [40]. We conducted all analyses in R version 3.6.0. We used the *spdep* package to run the spatial models [41] and the *metafor* package to conduct the meta-analyses [42].

## 3. Results

### 3.1. Descriptive Analyses

After removing tracts that were ineligible to gentrify, a total of 27,178 and 27,220 tracts from across 43 MSAs were included in the 1990–2000 and 2000–2010 analyses, respectively (Table 1). Among the gentrifiable tracts, median household income increased in 1990–2000 (mean increase: 2188 USD), but decreased in 2000–2010 (mean decrease: 1827 USD); percentage living in poverty remained the same in 1990–2000 (mean change: 0%) and increased in 2000–2010 (mean increase: 0.03%); median home value and median household rent increased in both time periods. There were small increases in percentage of the employed civilian population ages 16 and up working professional jobs (mean increase in 1990–2000: 0.07%; mean increase in 2000–2010: 0.02%) and with a bachelor’s degree (mean increase in both time periods: 0.04%). Meanwhile, in both time periods, there were decreases in percentage non-Hispanic white populations (mean decrease in 1990–2000: 0.10%; mean decrease in 2000–2010: 0.07%) and small increases in percentage non-Hispanic Black populations (mean increase in 1990–2000: 0.02%; mean increase in 2000–2010: 0.01%).

### 3.2. Preliminary Analyses: Associations between Percentage Greenspace and Any Gentrification (Categorical)

Table 2 presents meta-analytic estimates of the association between percentage greenspace and the categorical variable representing any gentrification. In unadjusted models, in 1992, census tracts with percentage greenspace greater than the 75th percentile of the distribution of percentage greenspace in their MSA in 1992 had 69% higher risk of gentrifying in 1990–2000 compared to those with percentage greenspace that was lower than the median for their MSA (95% CI: 1.45, 1.97). Also in unadjusted models, census tracts with >75th percentile of the distribution of percentage greenspace in their MSA in 2001 had a 23% higher risk of gentrifying in 2000–2010, compared to tracts with percentage greenspace cover lower than the 75th percentile (95% CI: 1.13, 1.34). After adjustment for population density in either 1990 or 2000, estimates were attenuated toward or to the null (1.23, 95% CI: 1.06, 1.42 for 1990–2000; 0.97, 95% CI: 0.90, 1.04 for 2000–2010).

### 3.3. Spatial Lag Models of Association between Greenspace and Sociodemographic or Housing Cost Changes

Although modest in magnitude, the direction of the unadjusted estimates of association from the spatial lag models was consistent with the hypothesis that higher percentage greenspace in either 1992 or 2001 was associated with subsequent interdecadal gentrification-related sociodemographic and housing cost changes. In unadjusted models, compared to census tracts with low percentage greenspace in 1992 (defined as ≤50th percentile of the MSA-specific distribution of percentage greenspace), in 1990–2000, census tracts with high percentage greenspace (defined as >75th percentile of the MSA-specific distribution) experienced decreases in percentages of the population who were non-Hispanic Black, Hispanic, and living in poverty, and increases in percentage of the population who were non-Hispanic White, percentage of adults with a bachelor’s degree, and percentage employed civilian population ages 16 and up working professional jobs. Relative to census tracts with the lowest percentage greenspace in 1992, tracts with high percentage greenspace also experienced modestly higher increases in rent, home value, and median household income (Figure 1a and Appendix A). Adjusting for population density in 1990 caused the estimates between percentage greenspace in 1992 and gentrification-related changes in 1990–2000 to move toward the null (Figure 1b). After adjusting for population density in 1990, compared to census tracts with low percentage greenspace in 1992, those with high percentage greenspace in 1992 experienced modestly higher increases in percentage of the employed civilian aged 16+ population working professional jobs in 1990–2000 (β: 0.18, 95% CI: 0.11, 0.26) and in median household income in 1990–2000 (β: 0.23, 95% CI: 0.15, 0.31). Although modest in magnitude and closer to the null, after adjusting for population density in 1990, we observed modest associations between higher percentage greenspace in 1992 and increases in 1990–2000 in median home value (β: 0.055, 95% CI: 0.002, 0.107), median household rent (β: 0.046, 95% CI: 0.012, 0.081), and percentage of the aged 25+ population with a bachelor’s degree (β: 0.072, 95% CI: −0.010, 0.154), as well as decreases in percentage Hispanic (β: −0.042, 95% CI: −0.073, −0.012).

Estimates of associations between percentage greenspace in 2001 and subsequent 2000–2010 sociodemographic and housing cost changes were similar in direction to those for the previous decade, but closer to the null (Figure 2 and Appendix A). The most substantial associations were for high vs. low percentage greenspace in 2001 (defined as >75th percentile vs. ≤75th percentile of the MSA-specific distribution in 2001) and 2000–2010 increases in percentage living in poverty, median household income, and percentage of the employed, civilian, aged 16+ population working professional jobs. In unadjusted models, compared to census tracts with low percentage greenspace in 2001, census tracts with high percentage greenspace experienced more substantial 2000–2010 decreases in percentage of the population living in poverty (β: −0.19, 95% CI: −0.25, −0.17), increases in percentage of employed adults working professional jobs (β: 0.15, 95% CI: 0.10, 0.19), and increases in median household income (β: 0.16, 95% CI: 0.11, 0.20). After adjusting for population density in 2000, nearly all estimates of association between percentage greenspace in 2001 and 2000–2010 gentrification-related sociodemographic or housing cost changes moved to the null, and, contrary to expectations, the association with change in percentage of the population with a Bachelor’s degree crossed the null, suggesting that high vs. low percentage greenspace tracts experienced more substantial decreases in percentage of the adult population with a bachelor’s degree (β: −0.095, 95% CI: −0.138, −0.051).

### 3.4. Heterogeneity

There was substantial heterogeneity of associations across MSAs, as evidenced by high I^2^ statistics. The I^2^ statistic for population-density adjusted estimates of association contrasting high vs. low percentage greenspace in 1992 (>75th percentile vs. ≤50th percentile of the MSA-specific distribution) and any gentrification in 1990–2000 was 78.70%, and that for the association contrasting high vs. low percentage greenspace in 2001 (>75th percentile vs. ≤75th percentile of the MSA-specific distribution) and any gentrification in 2000–2010 was 41.25% (Table 2). In the unadjusted analyses of associations between percentage greenspace in 1992 and changes in sociodemographic and housing price change variables in 1990–2000, the I^2^ statistic for the estimate of association contrasting high vs. low percentage greenspace (>75th percentile vs. ≤ 50th percentile of the MSA-specific distribution) ranged from 78.9% for change in percentage non-Hispanic Black, to 94.1% for change in percentage Hispanic (Appendix A). In unadjusted analyses of associations between percentage greenspace in 2001 and 2000–2010 sociodemographic and housing cost changes, the I^2^ statistic ranged from 60.6%, for change in percentage working professional jobs, to 85.8%, for change in median home value (Appendix A). Adjusting for population density reduced some of the heterogeneity. For example, for the 1990–2000 period, the I^2^ statistics for associations between percentage greenspace in 1992 and 1990–2000 changes in percentage non-Hispanic Black and percentage non-Hispanic White, contrasting high vs. low percentage greenspace, were reduced to 10.3% and 13.0%, respectively (Appendix A). The I^2^ statistics for associations between percentage greenspace in 2001 and 2000–2010 changes in percentage non-Hispanic Black and percentage non-Hispanic White, contrasting high vs. low percentage greenspace, were reduced to 2.8% and 56.7%, respectively (Appendix A).

Figure 3, Figure 4 and Figure 5 show forest plots of MSA-specific, population density-adjusted estimates of association between percentage greenspace and any gentrification (Figure 3), changes in median household income (Figure 4), and percentage of the ages 16+ civilian population working professional jobs (Figure 5) in 1990–2000 (a) and 2000–2010 (b). These figures illustrate heterogeneity across MSA, and they show that the magnitude and direction of associations with the neighborhood change variables were not consistent across the MSAs. In addition, these forest plots show that the outlier MSAs were not consistent across period. For example, the most substantial positive association between percentage greenspace in 1992 and any gentrification in 1990–2000 was in the metropolitan Louisville area. By contrast, there was a negative association between percentage greenspace in 2001 and any gentrification in 2000–2010 in Louisville.

### 3.5. Effect Measure Modification by Racial/Ethnic Composition

With two exceptions, we did not observe evidence of effect modification by race or ethnicity, as evidenced by the point estimates from one stratum being subsumed by the confidence limits of the other (Appendix A). In census tracts with higher percentage Hispanic populations in 1990, compared to census tracts with low percentage greenspace in 1992 (≤50th percentile for the MSA-specific distribution), census tracts with high percentage greenspace in 1992 (>75th percentile of the MSA- specific distribution) experienced more substantial 1990–2000 increases in percentage non-Hispanic white (β: 0.10, 95% CI: 0.05, 0.15) and more substantial 1990–2000 decreases in percentage Hispanic populations (β: −0.08, 95% CI: −0.12, −0.03); these associations were equal to null in lower percentage Hispanic tracts. In census tracts with high percentage non-Hispanic Black populations in 2000, compared to tracts with low percentage greenspace in 2001 (≤75th percentile for the MSA-specific distribution in 2001), those with high percentage greenspace in 2001 (>75th percentile for the MSA-specific distribution in 2001) experienced more substantial decreases in percentage of the civilian population ages 16 and over working professional jobs, 2000–2010 (β: −0.117, 95% CI: −0.17, −0.06). By contrast, this association was close to null in census tracts with lower percentage non-Hispanic Black populations.

### 3.6. Sensitivity Analyses

Results were similar when we defined neighbors using the rook rather than the queen contiguity matrix, although associations with changes in percentage with a bachelor’s degree, percentage working professional jobs, percentage living in poverty, and median household rent were slightly attenuated toward the null for the 1990–2000 time period (Appendix A). For example, the estimate of association between percentage greenspace in 1992 and 1990–2000 changes in percentage of the aged 25 and older population with a bachelor’s degree, contrasting high vs. low percentage greenspace census tracts, moved from β: 0.21 (95% CI: 0.15, 0.26) to β: 0.11 (95% CI: 0.05, 0.17). When we expanded the sample to include all census tracts, regardless of whether they were eligible to gentrify, associations remained similar. Lastly, when we used a more conservative criterion to define eligibility to gentrify (restricting to those below the MSA-specific 50th percentile for median household income), results were similar in direction, although some estimates moved closer to the null. For example, the estimate of association between percentage greenspace in 1992 and 1990–2000 changes in median household income moved from β: 0.35 (95% CI: 0.29, 0.41) to β: 0.26 (95% CI: 0.19, 0.32). The estimate of association between percentage greenspace in 2001 and 2000–2010 changes in percentage living in poverty moved from β: −0.191 (95% CI: −0.25, −0.13) to β: −0.12 (95% CI: −0.19, −0.04).

## 4. Discussion

In this spatial analysis of 43 metropolitan areas across the US, we observed modest associations between high percentage greenspace and subsequent sociodemographic and housing cost changes that are consistent with gentrification processes. After adjusting for census-tract level population density in 1990, areas with high percentage greenspace (defined as >75th percentile of the MSA-specific distribution) experienced higher 1990–2000 increases in percentage of the employed civilian aged 16+ population working professional jobs and in median household income; however, the overall associations were modest. The associations also differed in magnitude and direction across MSAs, suggesting that these trends are not uniform across cities.

This paper contributes to a growing body of scholarly research on environmental or green gentrification. Previous studies have found that investment in greenspace was associated with rising housing costs and sociodemographic changes consistent with gentrification [18]. In an analysis of ten major US cities, a higher proportion of census tracts that were within a half mile of a new greenway park gentrified compared to those located further distances [43]. In Brooklyn, New York (NY), the restoration of Prospect Park was associated with increases in median household income, loss of Black residents, and increased median home values in some neighborhoods surrounding the park [44]. Although green gentrification has been studied and documented more extensively in higher-income countries, research has also found disparities in access to greenspace [45] and evidence of green gentrification in lower- and middle-income countries [46,47,48]. Importantly, these green gentrification processes are not uniform within cities; recent studies examined how neighborhood and park characteristics modify the relationship between greenspace and gentrification [17,43,49,50]. Previous work has shown that factors such as proximity to other amenities shaped green-gentrification processes [17].

Within the context of this literature, our analysis makes two specific contributions. One is to demonstrate the importance of population density in analyses of green gentrification. Population density is often used as a proxy for urbanicity [51] and may also be correlated with urban amenities that attract gentrification. After adjustment for tract-level population density, our estimates of gentrification moved closer to or became equal to null. These results suggest that urbanicity and/or other urban amenities could partially explain the relationships between greenspace and sociodemographic shifts. Second, within the literature on green gentrification, our analysis is one of only a few to incorporate cross-city comparisons. While recent research by Rigolon and Nemeth analyzed data from ten US cities [43], most studies of green gentrification focus on a single city. Although our summary estimates suggested evidence of green-gentrification trends, we also observed substantial heterogeneity in associations between percentage greenspace and gentrification, in terms of both magnitude and direction, across the MSAs. Detailed exploration of the reasons for this heterogeneity is beyond the scope of the current analysis, but such research would be a fruitful next step. Overall, we observed little evidence of effect modification by race/ethnic neighborhood composition, although we observed an association between higher percentage greenspace and reductions in percentage Hispanic population in 1990–2000, only in census tracts with higher percentage Hispanic populations. Furthermore, we observed an association between high percentage greenspace and reductions in percentage of the population working professional jobs in 2000–2010, only in census tracts with high percentage non-Hispanic Black populations. Because these stratified estimates were very modest, they should be interpreted cautiously. However, these patterns may reflect segregated gentrification trends, consistent with implicit biases, stereotyping, and neighborhood stigmatization of racial minority neighborhoods by whites, which have been observed in past gentrification research [34,52]. Further research is needed to explore the complex relationship between neighborhood racial/ethnic composition and green gentrification.

Strengths of this analysis include the use of a large, national database spanning two decades. Our method of operationalizing gentrification was a variation of past gentrification census data-based measures. It was comparable to past measures in that we identified gentrifiable areas; however, alongside a categorical definition of gentrification, we also estimated the direction and magnitude of associations with changes in multiple sociodemographic and housing cost indicators. We used a validated, accepted, and relatively high-resolution greenspace measure to quantify percentage greenspace. We used spatial lag models to estimate MSA-specific associations; these models account for the lack of independence across census tracts. We also used a meta-analytic approach, which accounted for heterogeneity in associations between percentage greenspace and gentrification across the largest metropolitan areas in the US.

While the use of a large, national dataset is a strength, it also comes with corresponding limitations. Because this was a national-level analysis, the availability of data limited our ability to assess quality, type, perceived access, or function (e.g., active transportation, physical activity, hiking) of greenspace. Additionally, our greenspace measure does not distinguish previously existing greenspace from new greenspace production, nor does it account for recent renovations to existing greenspaces. These different characteristics of greenspace may affect associations with gentrification [43,49], and future research on relationships between greenspace and gentrification should seek to refine these measures. Another limitation is that the MRLC consortium changed its methodology for land-cover classification between 1992 and 2001—from an unsupervised and hybrid classification system to a decision tree method. Therefore, we were unable to directly compare changes in percentage greenspace between years 1992 and 2001. Furthermore, the quantitative estimates of association cannot and should not be used to make inferences about shifts or continuances of greenspace and gentrification associations over the two decades. In addition, due to data availability, we quantified associations over 10 year time spans. While this approach correctly acknowledges that gentrification is a multiyear process, it does not allow study of finer-scale temporal trends. While greenspace may be correlated with other urban amenities that attract gentrification [53] and may be part of larger redevelopment programs within urban areas [17], we did not examine proximity to amenities as a potential modifier or confounder of associations between greenspace and gentrification processes. This analysis included the 43 MSAs in the US with at least 300 census tracts. This criterion was established to ensure adequate statistical power for MSA-specific analyses and for stratification by racial/ethnic compositional categories. However, because of this criterion, our analysis may have excluded smaller MSAs in the US in which important gentrification processes occurred, and/or it may have included larger MSAs in which these processes were less prominent. Lastly, our census-based measures do not represent neighborhood perceptions of gentrification, nor do they capture the complex local politics, policy parameters, and rhetoric that surround urban greenspace development [15]. Given the nationwide scale of this analysis, it was not feasible to conduct interviews about neighborhood perceptions of gentrification and of perceived access to and value of green spaces. Future work using mixed methods would strengthen the inferences and improve interpretation of study findings.

Further research on the health implications of green gentrification for long-term neighborhood residents is needed. If green-gentrification processes contribute to the physical displacement of lower-income or racial/ethnic minority residents within neighborhoods, this means that, rather than benefitting them, the new, improved, or restored greenspaces contributed to severe cost, stress, and disruption among community members. Some studies have shown that displacement is not an inevitable outcome of gentrification [19,30,31]. Even if green gentrification does not lead to residential displacement, this process could have adverse impacts on the health of long-term residents that outweigh the benefits of greening the neighborhood [54]. For example, gentrification may lead to feelings of social and cultural exclusion among those who remain in the neighborhood [54]. Gentrification has been associated with poor health outcomes, including preterm birth among non-Hispanic Blacks [22] and diagnoses of anxiety or depression among children [23]. In Philadelphia, Pennsylvania (PA), gentrification was associated with improved self-rated health among neighborhood residents overall, but with poorer self-rated health among Blacks living in gentrifying neighborhoods [21]. Similarly, in California, residing in a gentrifying neighborhood was associated with improved self-rated health among residents who identified as white, but with poor self-rated health among those who identified as Black [55]. Particularly relevant to our analysis, a study from New York City found an association between greenspace and better self-rated health, but only among higher-income and more highly educated residents of gentrifying neighborhoods, suggesting differential perception of and/or access to greenspace within neighborhoods [14]. More research is needed to empirically document how greenspace shapes the health of low-income and racial/ethnic minority residents [13].

Policy discussion about ways to ensure equitable greening has centered around “just green enough” strategies, which take into account community interests and desires, rather than real estate and economic interests [20]. These strategies may also limit the scale or type of greening projects in order to benefit long-term residents [56,57]. More generally, anti-gentrification policies, such as rent stabilization or programs such as shared equity projects that give existing residents a stake in improving their neighborhoods, may support equitable greening [20]. There is a clear need for further research into ways to effectively prevent adverse gentrification consequences of urban greening. In addition, results from this analysis have implications for large, population-based epidemiologic studies that quantify links between greenspace and health endpoints. In particular, these results underscore the importance of evaluating and considering impacts of biases resulting from confounding by socioeconomic position or from self-selection of higher-socioeconomic-position residents into greener neighborhoods. However, our results also showcase the fact that relationships between greenspace and neighborhood changes may not be uniform across all places.

## 5. Conclusions

Recently, particularly in the wake of protests for racial justice, there is growing attention to the idea that marginalized city residents have too often been ignored in decision-making about environmental improvement and investment [58]. Results from our work suggest that, in major MSAs across the US, low-income census tracts with more greenspace experienced modest sociodemographic changes that are consistent with gentrification trends, underlining concerns about green gentrification. However, there was heterogeneity in these associations across areas. More research is needed to understand the role of urban policy regimes, social and economic contexts, and park and greenspace features in these cross-city differences, as well as to identify the implications for the health benefits of greenspace. The growing body of evidence about green gentrification indicates the importance of optimal greening strategies that prevent physical displacement and social exclusion of long-time residents and ensure that green infrastructure is accessible and beneficial for all.

## Figures and Tables

**Figure 1 ijerph-18-03315-f001:**
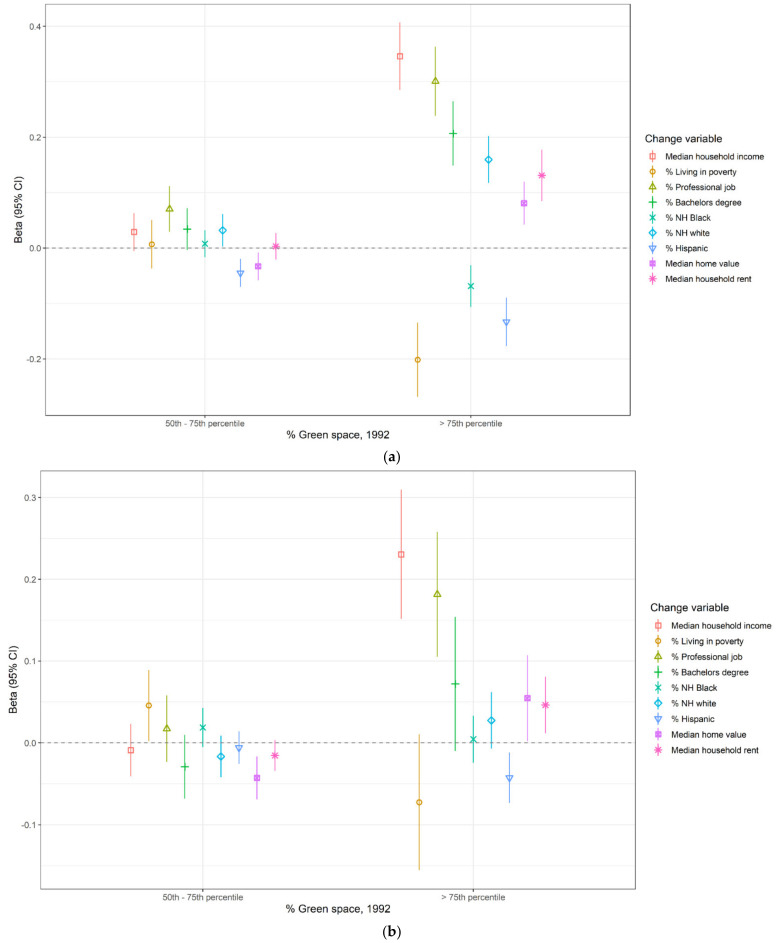
Meta-analytic estimates of association between percentage greenspace in 1992 and sociodemographic and housing cost changes in 1990–2000, among gentrifiable census tracts in the 43 largest MSAs in the United States: (**a**) unadjusted; (**b**) adjusted for population density in 1990.

**Figure 2 ijerph-18-03315-f002:**
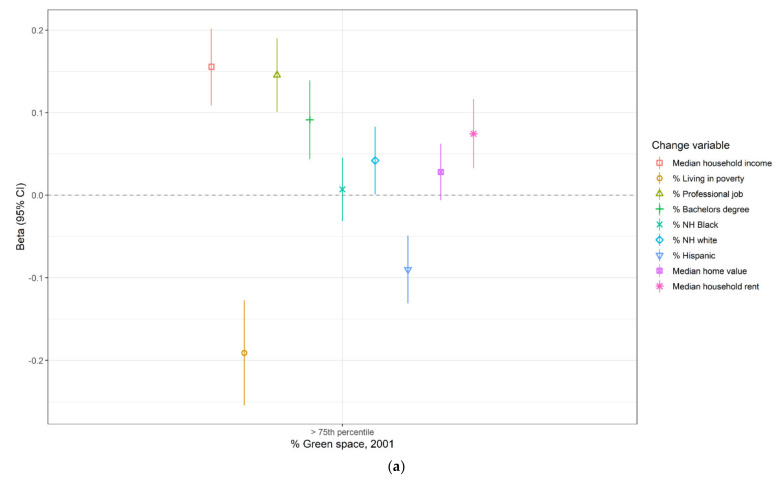
Meta-analytic estimates of association between percentage greenspace in 2001 and sociodemographic and housing cost changes in 2000–2010, among gentrifiable census tracts in the 43 largest MSAs in the United States: (**a**) unadjusted; (**b**) adjusted for population density in 2000.

**Figure 3 ijerph-18-03315-f003:**
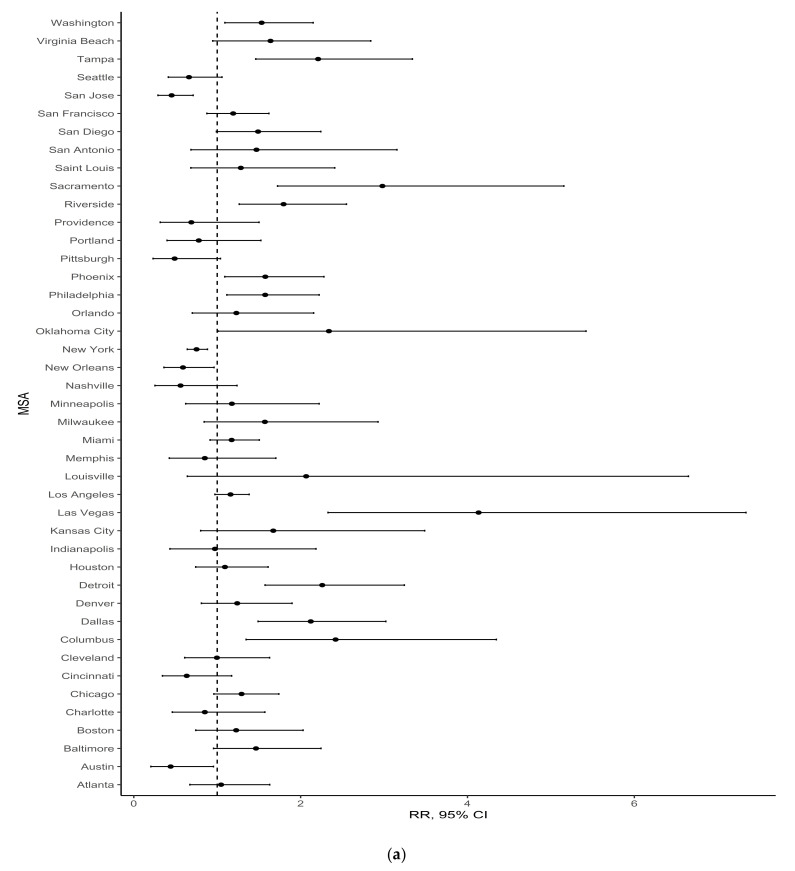
MSA-specific risk ratio (RR) and 95% confidence interval (CI) estimates of association between percentage greenspace and any gentrification, adjusted for population density, for the periods (**a**) 1990–2000 or (**b**) 2000–2010. Estimates for the 1990–2000 period represent comparisons of census tracts with percentage greenspace >75th percentile vs. ≤50th percentile for their MSA. Estimates for the 2000–2010 period represent comparisons of census tracts with percentage greenspace >75th percentile vs. ≤75th percentile for their MSA.

**Figure 4 ijerph-18-03315-f004:**
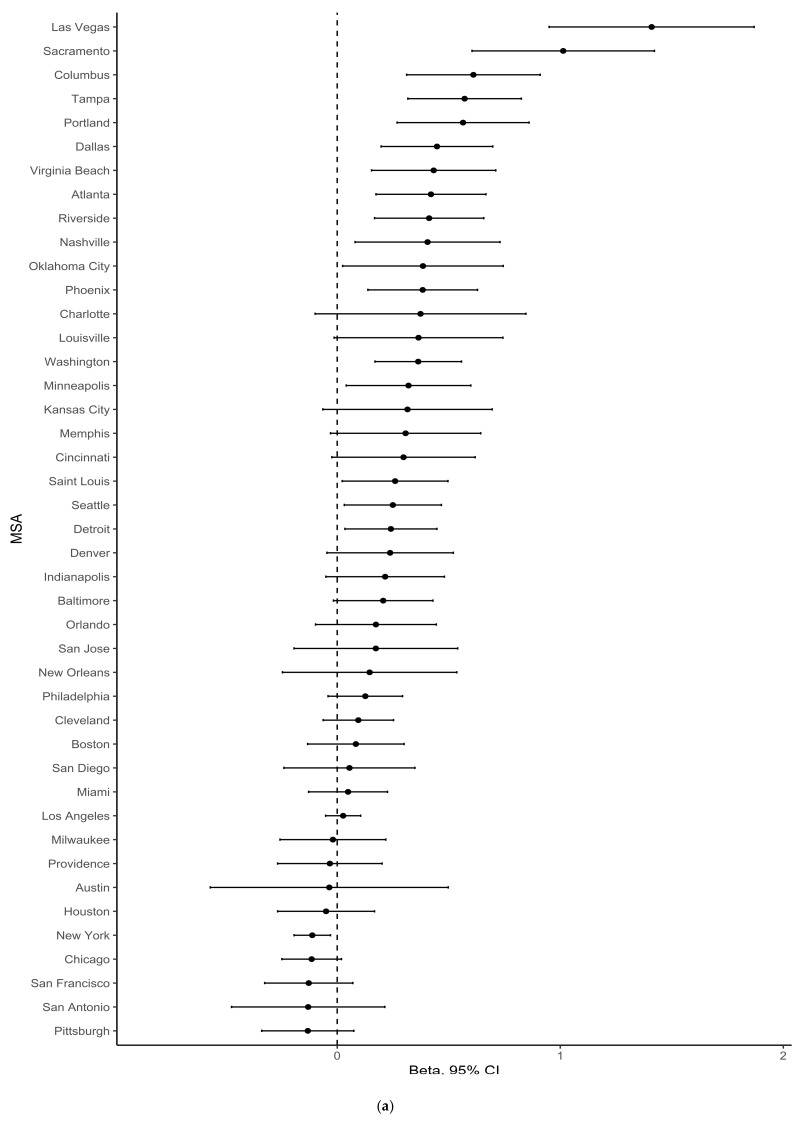
MSA-specific estimates of association between percentage greenspace and changes in household income, adjusted for population density, for the periods (**a**) 1990–2000 or (**b**) 2000–2010. Estimates for the 1990–2000 period represent comparisons of census tracts with percentage greenspace > 75th percentile vs. ≤50th percentile for their MSA. Estimates for the 2000–2010 period represent comparisons of census tracts with percentage greenspace > 75th percentile vs. ≤75th percentile for their MSA.

**Figure 5 ijerph-18-03315-f005:**
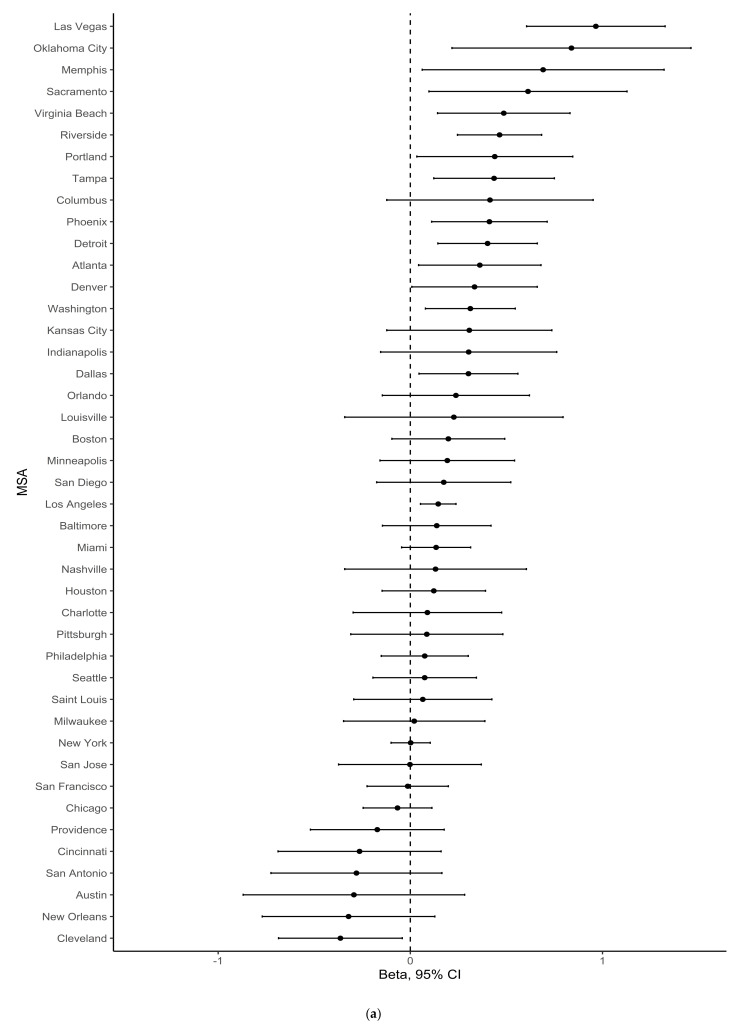
MSA-specific estimates of association between percentage greenspace and changes in percentage of the civilian population ages 16 and over working professional jobs, for the periods (**a**) 1990–2000 or (**b**) 2000–2010. Estimates for the 1990–2000 period represent comparisons of census tracts with percentage greenspace > 75th percentile vs. ≤50th percentile for their MSA. Estimates for the 2000–2010 period represent comparisons of census tracts with percentage greenspace > 75th percentile vs. ≤75th percentile for their MSA.

**Table 1 ijerph-18-03315-t001:** Mean change in sociodemographic and housing cost variables, 1999–2000 and 2000–2010, across the gentrifiable census tract.

	Mean (SD)	
	1990–2000	2000–2010
Total gentrifiable census tracts	27,178	27,220
Median household income (USD)	2188.07 (10654.28)	−1826.69 (12344.70)
% living in poverty	0.00 (0.06)	0.03 (0.08)
Median home value (USD)	27,092.31 (57,844.46)	107,330.20 (113,345.34)
Median household rent (USD)	18.43 (145.16)	110.92 (199.20)
% working professional jobs	0.07 (0.07)	0.02 (0.09)
% bachelor’s degree	0.04 (0.07)	0.04 (0.08)
% Non Hispanic white	−0.10 (0.11)	−0.07 (0.10)
% Non Hispanic Black	0.02 (0.08)	0.01 (0.07)
% Hispanic	0.05 (0.08)	0.04 (0.08)

**Table 2 ijerph-18-03315-t002:** Relative Risk (RR) and 95% Confidence Interval (CI) estimates of association between percentage greenspace and any gentrification, from 1990–2000 or 2000–2010 ^1,2^.

	≤50 Percentile	>50th–75th Percentiles				>75th Percentile			
	RR	RR	95% CI			RR	95% CI		
			LL	UL	I^2^		LL	UL	I^2^
Any gentrification, 1990–2000									
Unadjusted	Ref	1.19	1.08	1.32	79.62	1.69	1.45	1.97	93.48
Adjusted for population density	Ref	1.01	0.93	1.09	49.84	1.23	1.06	1.42	78.70
Any gentrification, 2000–2010									
Unadjusted	Ref.	Ref.				1.23	1.13	1.34	83.95
Adjusted for population density	Ref.	Ref.				0.97	0.90	1.04	41.25

Abbreviations: Relative Risk (RR); CI, Confidence Interval; LL, Lower Limit; UL, Upper Limit. ^1^ The reference categories for the 1990–2000 and 2000–2010 periods are census tracts with ≤50th and ≤75th percentiles, respectively, of the distribution greenspace across census tracts in their metropolitan statistical area (MSA). ^2^ The I^2^ statistic describes the percentage of variation across Metropolitan Statistical Areas that is due to heterogeneity rather than chance.

## Data Availability

All data used from this analysis are publicly available.

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
