# Peer review of "Associations between Greenspace and Gentrification-Related Sociodemographic and Housing Cost Changes in Major Metropolitan Areas across the United States"

_ijerph, 2021, doi:10.3390/ijerph18063315_

Round 1

Reviewer 1 Report

#Comments to authors

The manuscript, entitled “Associations between greenspace and gentrification-related sociodemographic and housing cost​changes in major metropolitan areas across the​​​ United States, ” aimed to quantify associations between baseline greenspace (in 1992 and 2001) and subsequent neighborhood compositional change (from 1990-2000, and 2000-2010), and to quantify heterogeneity in these relationships across major U.S. cities. This is an interesting study corresponding to the problems of many countries, particularly developing countries, where urban inequity is inevitably occurring. Gated communities and condos are being developed in many cities in developing countries, and these types of buildings are for middle and high income people. The poor and other ethnic groups are being neglected or perceive excluded.

  • Introduction
    • Literature in the introduction was not strong. We could see an explanation of the benefits of greenspace in the 1st and 2nd paragraphs and an explanation of gentrification in the third and fourth paragraphs. However, the explanation was not strong enough to convince readers of what has been done in related areas, particularly gentrification. Further review of literature on gentrification in the US cities and other cities should be added.
    • The purposes of the study were clearly stated.
  • Results:
    • 3,4 and 5 are not clear. Please provide higher resolution figures.
  • Discussion
    • It was clear that the authors tried to explain the results and compared them with other studies in other states of the US. It would be more interesting to add some studies investigating a relationship between greenspace and gentrification in developing countries or the least developed countries.
  • Conclusion
    • It would be great if the authors restated some key policy implications of the study.

Reviewer 2 Report

The manuscript analyses the association between greenspace and subsequent gentrification in the United States. It is an interesting and well-structured paper, and provides good methodological and phenomenological contributions to studies focused on gentrification in urban areas. However, some topics should be addressed before it can be considered for publication.

2. Materials and Methods

  • As the periods 1990-2000 and 2000-2010 don't have any specific association with the research object and were chosen due to methodological limitations (availability of census data), this should be very briefly clarified in the first paragraph of section 2.1.
  • Section 2.2 is repeating some information already presented in the previous section. This section should be restructured in order to aggregate some information presented in section 2.1 (the explanation on MSAs and the chosen time periods, for instance), or removed from the paper. In the second case, the number of census tracts (which is the only novelty presented in section 2.2) might be added to the second paragraph of section 2.1.
  • The methods of landcover assessments changed between 1992 and 2001, results for period 1990-2000 and 2000-2010 and they are indeed internally consistent, but they might be overestimating or underestimating the estimated effect of greenspace on subsequent gentrification in different ways in those two periods. This makes it particularly hard for the authors to provide more insightful contributions about possible shifts or continuances of gentrification processes in the US. One example is how we interpret the information that in 1992, urban areas in the 75th percentile of greenspace had a 69% higher risk of gentrifying, while in 2000, this risk dropped to 23%. If the method to identify the dependent variable changed, how do we analyse these numbers together? Did the US upper classes valued less greenspace in the 00s than they did in 90s, and therefore held back some gentrification processes? If so, which social, cultural and environmental changes might be related to this shift? The methodology used in the paper -- if it weren't for the change in the methods of greenspace identification -- would be able to provide quantitative data to help answering those questions, which would make the paper much stronger. 
  • Note that the substantive effect of the methodological changes in the NLCD on how we look at the results of the paper might be reduced by a better understanding of what actually changed. A brief explanation of these methodological changes should be added to the last paragraph of section 2.3.1.
  • Moran's I statistics should be added to the supplementary file.
  • The study does not add any qualitative attribute to greenspaces, though they change substantially in urban areas. Parks and squares provide a much different level of attraction to new residents than suburban lands. 

3. Results 

  • Results are based not exactly in places where gentrification occurred (socio-demographic variables changed more than the average), but in places where gentrification most likely occurred. Gentrification might have happened in places which did not meet the cut and gentrification might not have happened in places above the cut. This should be clearer in the paper.
  • Some results lack a deeper investigation or explanation, and should be related to the literature. 
    • Contrary to the other models, for example, the % of black population showed a positive association with greenspaces in the adjusted model for the period 2000-2010, but this is not mentioned in the text.
  • Figure 3 (a) and (b) should have the same scale. Heterogeneity seem to be much higher in 1990-2000 than in 2000-2010, but the graphs suggest otherwise.

4. Conclusions

The last section of the paper is the weakest one. More than presenting the limitations of the study, the authors should summarize their findings and relate them to the literature on gentrification and greenspaces.

Reviewer 3 Report

The article addresses green spaces' attractiveness capacity in the USA's 43 largest metropolitan statistical areas (MSA). The analysis is mostly quantitative, based in statistics and common indicators offered by the USA official entities. From the article, it does not seem clear that the authors approached the possibility of conceiving new indicators, thus resulting in rather superficial conclusions.

While correct, the reference system is complicated to follow; Harvard or APA would fit best, even if more used in SSH.

While the title and abstract clearly state the article discusses the correlations between greenspace and gentrification-related with sociodemographic and housing affordability, the article background (in line with the authors' backgrounds, seemingly) starts by approaching the health values the proximity of greenspaces.

The article intends to address gentrification. Although mentioning various types (which are not consensual and depend on the literature, is context-dependent even within the USA), it fails to define gentrification within the approach scope. The social dimension is somewhat elementary and does not provide robust support to the authors' claims; this is especially visible in the list of references.

The methods do not the article's aim, or the article's topic should meet the methods used, which are just quantitative. In such cases, methods should be mixed and include both quantitative and qualitative data. Hence, the discussion and conclusions result as descriptive and based on the possibility of "measuring" gentrification.

Round 2

Reviewer 2 Report

Authors have responded to all referees' comments. Although no methodological change has been made from the previous version (which would make the manuscript stronger), limitations, results and conclusions are now better described. 

Reviewer 3 Report

The reviewer acknowledges the authors' efforts to clarify and improve the article in such an adequate way